# Incineration Kinetic Analysis of Upstream Oily Sludge and Sectionalized Modeling in Differential/Integral Method

**DOI:** 10.3390/ijerph16030384

**Published:** 2019-01-29

**Authors:** Yanqing Zhang, Xiaohui Wang, Yuanfeng Qi, Fei Xi

**Affiliations:** 1School of Environmental and Municipal Engineering, Qingdao University of Technology, Qingdao 266033, China; zyq_luck@163.com (Y.Z.); vadaxi@163.com (F.X.); 2Technical Test Center of Shengli Oil Field, Dongying 257001, China; wxhui583@126.com

**Keywords:** oily sludge, incineration, kinetic, modeling, differential/integral method

## Abstract

As the most significant solid residue generated in the oil production industry, upstream oily sludge was regarded as hazardous waste in China due to its toxicity and ignitability, and to date, the incineration process has been considered the most efficient method in practice. Due to the complicated components of oily sludge, a kinetic model of the incineration process was difficult to build, and is still absent in engineering use. In this study, multiple non-isothermal thermogravimetric analysis (TGA) and differential scanning calorimetry (DSC) analysis were applied for the kinetic analysis of upstream oily sludge in air conditions. A viewpoint regarding the rules to sectionalize the reaction stages was raised, and a differential integral method to obtain the incineration kinetic model was provided. The results showed that four stages that were divided based on the weight-loss regions in the TGA curves and the endothermic/exothermic sections in the DSC curves were suitable to obtain an incineration kinetic model of oily sludge. The integral method was beneficial for obtaining the average activation energy of each stage, and the differential method was suitable for gaining the nth-order reaction rate equation and the pre-exponential factor before the operating temperature became lower than 635.968 °C. The average activation energies of stages one, two, three, and four were 60.87 KJ/mol, 78.11 KJ/mol, 98.82 KJ/mol, and 15.96 KJ/mol, respectively. The nth-order reaction rate equations and pre-exponential factors of stages one, two, and three were 0.82, 3.50, and 2.50, and $e^{13.32}\min^{-1}$, $e^{19.69}\min^{-1}$, and $e^{21.00}\min^{-1}$, respectively.

## 1. Introduction

Upstream oily sludge is the most significant solid waste generated in the oil production industry, and is mainly discharged from the crude oil storage process [1,2]. Ordinarily, crude oil is housed in oil tanks prior to being refined to petroleum products, and the heavier species are separated and settled at the bottom of the storage tanks [3,4,5]. The solid sediments are the major components of upstream oily sludge, which contains a high concentration of complex petroleum hydrocarbons (PHCs, e.g., asphaltenes, resins, and tar), fine solids, and heavy metals [6,7]. On account of the toxicity and ignitability characteristics, which represented a significant adverse effect to ecosystem and human health, both upstream and downstream oily sludge have been regarded as hazardous waste in China since 2008 [8,9,10]. A variety of oil recovery and/or sludge disposal methods have been studied for the treatment of upstream oily sludge, such as thermal treatment (incineration or pyrolysis) [11,12,13], solidification [14], solvent extraction [15,16], photocatalysis [17], ultrasonic treatment [18], and biodegradation [18,19,20,21]. In China, the incineration process was identified as the most efficient method for the disposal of upstream oily sludge, and has been successfully designed, established, and commercialized in the last few years. However, the other mentioned methods have been rarely applied in practice for failing to reach a compromised balance between satisfying the strict environmental regulations and maintaining a reasonable operating cost [22,23]. Various incinerators such as circulating fluidized bed combustion, rotary kiln, and chain boiler combustion were adopted in the industrial application and operated with a combustion temperature between 730–1200 °C [1,12,23]. Furthermore, excess air and auxiliary fuels were indispensable for the incineration process. The incineration product was directly affected by a variety of factors, including the pretreatment method, operating temperature, residence time, feedstock quality, and addition of auxiliary fuels [24].

Most of the current studies focused on the thermal co-treatment of upstream oily sludge with auxiliary solid waste and/or the by-products that exist in gaseous phases and solid residue [22,25,26]. Generally, thermal analysis occupied the pivotal position throughout the thermal treatment of solid waste, and was frequently studied in the dehydration, carbonization, and incineration of industrial waste such as red mud, sewage sludge, and antibiotic residues [27,28,29]. The thermogravimetric analysis (TGA) test was a representative non-isothermal method for thermal kinetics analysis that was sensitive enough to exhibit the weight loss of the reactant with the operating temperature/time. It was usually applied for the thermal decomposition of certain reactants in air or nitrogen conditions. Ordinarily, the reaction kinetics of thermal decomposition was represented by the nth-order reaction rate equation [30,31,32,33] (see Equations (1)–(3)). Furthermore, the Arrhenius equation (Equation (4)) was commonly utilized to describe the reaction rate constant.
(1)dα/dt=K(T)f(α)
(2)$$\alpha =\left(W_{i}-W_{t}/W_{i}-W_{e}\right)\times 100\%,\;\alpha \in \left(0\%-100\%\right)$$
(3)$$f\left(\alpha \right)={\left(1-\alpha \right)}^{n}$$
(4)$$K\left(T\right)=Aexp\left(-E_{a}/RT\right)$$
where *t* and *T* are the operating time and temperature; α is the conversion ratios of the reactant; and dα/dt is the relationship between the instantaneous conversion ratio and the operating time. In Equation (2), $W_{i}$, $W_{n}$, and $W_{e}$ are the initial weight, weight at a certain time, and the final weight of the reaction, respectively. In Equations (1) and (4), K(T) is the reaction rate constant. In Equation (3), f(α) is the nth-order reaction rate equation, and n is the reaction order. Meanwhile, in Equation (4), A is the pre-exponential factor; $E_{a}$ is the activation energy; and *R* is the gas constant.

The main objective for thermal kinetics analysis was to obtain the basic three elements [33], i.e., the activation energy ($E_{a}$), the pre-exponential factor (A), and the representation of the nth-order reaction rate equation (f(α)). Obviously, it was imprecise to distinguish the mass signal versus time or temperature in a single isothermal or non-isothermal thermogravimetric test. Thus, multiple non-isothermal thermogravimetric analyses were often applied for the thermal kinetic studies. If the relationships between the operating temperature and reaction time were in the form of Equation (5), and meanwhile, the heating rate was constant in a certain TGA test, Equation (1) could be re-written as Equation (6):(5)$$T=\beta t+T_{o}\Rightarrow dT/dt=\beta$$
(6)$$d\alpha /dT=A/\beta \times exp\left(-E_{a}/RT\right)f\left(\alpha \right)$$
where T and $T_{o}$ are the operating temperature and initial temperature, respectively; β is the heating rate, dα/dT is the relationships between the instantaneous conversion ratios and the operating temperature.

Equation (6) was the basic differential form for the study of thermal kinetic analysis, which represented the relationships between the instantaneous conversion ratios of the reactant with *T* under certain heating rates (β). The Friedman method [34] and Coats–Redfern method [35] were obtained by rearranging Equation (6) and applied to the thermal kinetic analysis in the pyrolysis process of oily sludge, effectively [12]. However, limitations for differential methods still existed and were mainly attributed to dα/dT, which was dramatically affected by the background noise of the TGA test [36,37,38,39,40]. Therefore, the activation energy ($E_{a}$) obtained in differential methods was imprecise.

Based on Equation (6), the integral method for the study of thermal kinetic analysis could be deduced as follows:(7)$$d\alpha /dT=\frac{A}{\beta }exp\left(-E_{a}/RT\right)f\left(\alpha \right)\Rightarrow \frac{1}{f\left(\alpha \right)}d\alpha =\frac{A}{\beta }e^{-E_{a}/RT}dT$$
(8)$$G\left(\alpha \right)=\int_{0}^{1}\frac{1}{f\left(\alpha \right)}d\alpha =\frac{A}{\beta }\int_{T_{o}}^{T}e^{-E_{a}/RT}dT$$
where *G*(α) is the integral Equation of f(α); and $T_{o}$ and *T* are the initial and final operating temperature, respectively.

The activation energy ($E_{a}$) and pre-exponential factor (A) could be obtained by rearranging Equation (8), and the negative effect of background noise could be avoided. However, the solution of the nth-order reaction rate equation f(α) and the reaction order (n) were hard to obtain. Therefore, both the differential method and integral method have their advantages and limitations during the acquisition of the activation energy, pre-exponential factor, and the nth-order reaction rate equation.

The pyrolysis kinetics analysis of oily sludge or plastic was reported in previous studies and the methods that were used are listed in Table 1, including the utilized thermal test method, the modeling method, and the basic three elements ($E_{a}$, A, and f(α) or *n*).

The reaction schemes of upstream oily sludge incineration were extremely complex due to the complicated composition; meanwhile, the reaction mechanism and the corresponding kinetic parameters for the incineration of various intermediate products and by-products may differ with the change of heating rate and operating temperature regions. It is difficult to identify or distinguish whether or not the kinetics model is suitable for the different reaction stages during the incineration process only on the basis of TGA curves. However, few studies have been available concerning the incineration reaction kinetics or in both differential and integral modeling methods for upstream oily sludge.

The aims of the present work were as follows. (1) Both the multiple non-isothermal thermogravimetric analysis (TGA) and differential scanning calorimetry (DSC) were performed to study the incineration thermal kinetic of oily sludge, simultaneously. (2) The work aimed to provide a new viewpoint to sectionalize the reaction stages in TGA/DSC curves. (3) The work aimed to present and utilize a comprehensive differential integral method to obtain the incineration kinetics model in different reaction stages.

## 2. Materials and Methods

### 2.1. Materials and Reagents

Purified air for DSC/TGA analysis was purchased from Qingdao Fengtai Co., Ltd.(Qingdao, China). Oxygen gas for the heating value test, with 99.99% purity, was purchased from Qingdao Chunfeng Co., Ltd.(Qingdao, China). The upstream oily sludge that was utilized in this study was obtained from a temporary storage bin in Shengli Oil Field, Dongying, Shandong province, China, and the samples appear to be viscous and black block.

### 2.2. Apparatus and Methods

The upstream oil sludge was dried in a recycle ventilation drier for 24 h at 105 °C; then, the heating value and ash content of the dried sample was analyzed by an automatic oxygen bomb calorimeter (SDAC6000, Sunday, Changsha, China) and automatic ash Fusion Tester (SDAF105b, Sunday, Changsha, China), respectively. The test of bulk density and moisture content for the undried upstream oily sludge was the same as that in previous studies [27,29].

First, seven to 15 mg of undried upstream oily sludge was employed for the TGA/DSC test and carried out in a SDT Q600 thermal analyzer (TA instrument, New Castle, PA, USA) under air atmosphere. In order to simulate the oxygen-rich conditions applied in practical incineration processes, ratios of the purified air to the initial weight of oily sludge were maintained over 10.00 mL/min to 1.00 mg. The operating temperatures for the TGA/DSC test were performed from 30 ± 2.5 °C to 900 °C. The heating rates ($\beta_{n}$) were five K/min, 10 K/min, 15 K/min, 20 K/min, and 25 K/min, and were labeled as $\beta_{a}$, $\beta_{b}$, $\beta_{c}$, $\beta_{d}$, and $\beta_{e}$, respectively. The sample of each experiment was weighted within a thousandth of an error and loaded into the quartz disk, which settled in the center of the equipment. A K-type thermocouple was inserted beside the quartz disk for measuring the operating temperature. Before the formal thermal test, the purified air controlled by a rotameter was injected into the equipment and lasted at least two hours for the purpose of purging. When the run was finished, the air was kept flowing until the temperature of the system returned to room temperature. The TGA and DSC data were simultaneously recorded, and the results of each experiment were repeated twice and averaged.

## 3. Results and Discussion

### 3.1. Sectionalized Rules and Peak-Thermal Kinetic Analysis

#### 3.1.1. Characteristics of the Upstream Oily Sludge

Features of the upstream oily sludge are represented in Table 2. Compared with the characteristics of the oily sludge, which was studied by Jing [8] and Xu [9], the upstream oily sludge had higher ash content and heating values, but lower moisture, which was attributed to the quality of the crude oil and the additives that were utilized in the recovery and dehydration processes.

#### 3.1.2. Sectionalized Rules for the Incineration Process of Upstream Oily Sludge

The results of TGA/DSC tests under oxygen-rich conditions for upstream oily sludge are shown in Figure 1 for when the heating rates of *βn* were $\beta_{a}$ = 5 K/min (Figure 1A), $\beta_{b}=$ 10 K/min (Figure 1B), $\beta_{c}=$ 15 K/min (Figure 1C), $\beta_{c}=$ 20 K/min (Figure 1D), and $\beta_{d}=$ 25 K/min (Figure 1E).

As shown in the five TGA curves (Figure 1A–E), the weight loss of upstream oily sludge in the incineration process was 45–47%, and three declining regions were simultaneous obtained. The first weight loss region was obtained from 30 °C to 280–350 °C, and the weight loss was 15–20%. The second weight loss region of 25–30% was obtained with the operating temperature ranging from 280–350 °C to 461.45–553.637 °C. The third weight loss region started at 589.248–630.992 °C with a slight weight loss of 1.5–5%.

As it was shown in the five DSC curves (Figure 1A–E), two significant exothermic reactions were detected at 237.905–391.764 °C and 341.465–553.637 °C, sequentially. It was the same as reported in the TGA/DTG test for the pre-dried oily sludge in nitrogen atmosphere [11,12]. Je-Lueng [12] considered that the former exothermic reactions in the pyrolysis process were attributed to the volatilization of volatile contents such as combined water, small hydrocarbons, and small molecular acids, while the latter exothermic reactions were caused by the decomposition of macromolecular compounds, such as for instance, tar, aromatic hydrocarbons, and cycloalkanes. Based on the above-mentioned studies, we inferred that the continuous exothermic reactions in the incineration process were possibly attributed to the volatilization and/or combustion of the volatile contents (in lower temperature region) and macromolecular compounds (in the higher temperature region), respectively. Although the components of volatile contents and macromolecular compounds were not the main objective of this work, the kinetics of each exothermic reaction seemed to be quite different. When the operating temperature exceeded 600 °C, with the increase of the heating rate, an endothermic phenomenon was gradually detected, which can probably be attributed to the decomposition of inorganic carbonate [27,28].

When the operating temperature was lower than 237.905 °C (Figure 1A–E), no exothermic or endothermic phenomenon occurred in the DSC curves, but a significant weight loss was observed in the TGA curves. When the exothermic or endothermic phenomenon appeared, the weight loss simultaneously accelerated. Both the endothermic and exothermic peak simultaneously shifted to the right with the increase of heating rate from βa to βe, and it was more sensitive and conspicuous in the exothermic regions.

Different types of volatilization decomposition and/or combustion reactions probably occurred in different weight loss regions, which were attributed to the complex components that existed in the upstream oily sludge. Therefore, both the mechanism and kinetic model changed with the increase of operating temperature. Based on whether the endothermic and/or exothermic reactions occurred (or not) in DSC tests, the TGA curves could be sectionalized as four weight loss stages; these are shown in Figure 2A.

As it was shown in Figure 1A, the ending temperature was 511.05–550.93 K in Stage 1, and no endothermic or exothermic reaction was detected, but a 10% weight loss ratio was obtained. The first and second exothermic reaction occurred in Stage 2 (weight loss = 10%) and Stage 3 (weight loss = 20%), and the ending temperatures were 614.62–664.91 K and 719.15–822.69 K, respectively. As shown in Figure 1B–E, the weight loss in the TGA curve caused by the endothermic phenomenon started at 614.422 ℃, 620.927 ℃, 627.884 ℃, and 630.992 ℃, respectively. Meanwhile, the DSC curve of each $\beta_{n}$ (10 K/min, 15 K/min, 20 K/min, and 10 K/min) showed that the endothermic temperature regions were 635.968–684.208 ℃, 655.017–726.35 ℃, 659.994–776.673 ℃, and 664.713–759.575 ℃, respectively. Therefore, the temperature regions for the incineration kinetic modeling of Stage 4 should be set as 635.968–900.00 ℃. The thermal parameters such as the weight loss ratios and peak temperature at βn (*n* = a, b, c, d, and e) were simultaneously obtained in Table 3.

It was not the weight loss, but rather the conversion ratios of the reactant that were employed in the equations of the incineration kinetics model. Therefore, the instantaneous weight of the samples detected in the TGA curves (Figure 1A–E) could be rearranged as instantaneous conversion ratios at certain operating temperature. The results of the conversion ratios versus operating temperature of stages one through four are shown in Figure 2B–E, respectively. In addition, the relationship between the instantaneous conversion ratio and the operating temperature (dα/dT) was the slope or the first-order derivative of each curve, as shown in Figure 2B–E. For instance, when the conversion rate was 55% and the heating rate was five K/min (in Stage 1), $d\alpha_{55\%}/dT$ equaled the slope (K), which is shown in the enlarged area of Figure 2B.

#### 3.1.3. Peak-Thermal Kinetic Analysis of Endothermic/Exothermic Reactions

Based on Equations (1), (3), and (4), the basic differential form for the study of thermal kinetic analysis could also be represented as Equation (9).
(9)$$d\alpha /dt=Ae^{-E_{a}/RT}\times {\left(1-\alpha \right)}^{n}$$
taking the quadratic differential on both sides of Equation (9), the following equations were obtained:(10)$$\begin{array}{cc} \frac{d}{dt}\left[\frac{d\alpha }{dt}\right] & =A{\left(1-\alpha \right)}^{n}\frac{de^{-E_{a}/RT}}{dt}+Ae^{-E_{a}/RT}\frac{d{\left(1-\alpha \right)}^{n}}{dt}\\ & =\frac{d\alpha }{dt}\times \frac{E_{a}}{{RT}^{2}}\times \frac{dT}{dt}-{Ae}^{-E_{a}/RT}\times n{\left(1-\alpha \right)}^{n-1}\frac{d\alpha }{dt}\\ & =\frac{d\alpha }{dt}\left[\beta E_{a}/RT^{2}-Ae^{-E_{a}/RT}\times n{\left(1-\alpha \right)}^{n-1}\right] \end{array}$$
when the quadratic differential $\frac{d}{dt}\left[\frac{d\alpha }{dt}\right]$ equals zero, which means the maximum or minimum value could be obtained. In DSC curves, the exothermal peak was the maximum value, and the endothermic peak was in response to the minimum value. At the limit value, Equation (10) equals zero, and the peak thermal kinetic equation was expressed as Equation (11):(11)$$\beta E_{p}/RT_{p}{}^{2}=Ae^{-E_{p}/RT_{p}}\times n{\left(1-\alpha_{p}\right)}^{n-1}$$
where $E_{p}$ and $T_{p}$ are the exothermal/endothermic peak activation energy and exothermal/endothermic peak operating temperature; and $\alpha_{p}$ is the exothermal/endothermic peak conversion ratio of the reactant;

Kissinger [36] considered that the formula $n{\left(1-\alpha_{p}\right)}^{n-1}$ equaled one, taking the natural logarithm of Equation (11) and rewriting it as Equation (12):(12)$$ln\frac{\beta }{T_{p}^{2}}≅ln\frac{AR}{E_{p}}-\frac{E_{p}}{RT_{p}};\;ln\frac{\beta_{n}}{T_{n-p}^{2}}\sim \frac{1}{T_{n-p}}$$
where $T_{n-p}$ was the peak temperature obtained in the DSC curve, and changed with $\beta_{n}$.

A straight line with slope $-E_{p}/R$ could be obtained by plotting $ln\frac{\beta_{n}}{T_{n-p}^{2}}$ versus $\frac{1}{T_{n-p}}$ at every endothermic or exothermic peak parameter. This method was called as “Kissinger approach” in this study. The results of the linear fittings for the endothermic peak and the two exothermic peaks were shown in Figure 1F and Table 2. The peak activation value ($E_{p}$) and the coefficient of determination (R^2^) of stages two, three, and four were 64.43 ± 5.34 KJ/mol (0.9798), 90.71 ± 13.35 KJ/mol (0.9389), and 102.11 ± 28.93 KJ/mol (0.8616), respectively.

### 3.2. The Reasoning Process of the Modeling Method Applied for Oily Sludge Incineration

#### 3.2.1. The Reasoning Process of Differential Methods

By rearranging and taking the natural logarithm in Equation (6), the activation value obtained in the differential method was shown in Equation (13). Friedman [37] considered that the activation value could be solved in spite of both the nth-order reaction rate equation and the pre-exponential factor. At each heating rate, a straight line with slope $-E_{a}/R$ could be obtained by plotting $ln\beta_{n}\left(d\alpha /dT\right)$ versus $\frac{1}{T}$:(13)$$\beta \times d\alpha /dT=Aexp\left(-E_{a}/RT\right)f\left(\alpha \right)\Rightarrow ln\beta \left(d\alpha /dT\right)=lnA+lnf\left(\alpha \right)-E_{a}/RT$$

If the activation value had been solved and the nth-order reaction rate equation was fitted to Equation (3), simultaneously, the intercept (lnA+lnf(α)) of the straight lines could be applied to solve the reaction order (*n*) and pre-exponential factor (A) from the equation in two unknowns established under different heating rates. Based on Friedman methods (Equation 13), another pathway to gain the reaction order (*n*) and the pre-exponential factor (A) was shown in Equation (14):(14)$$\beta \times d\alpha /dT=Aexp\left(-E_{a}/RT\right)f\left(\alpha \right)\Rightarrow ln\frac{\beta \left(d\alpha /dT\right)}{exp\left(-E_{a}/RT\right)}=lnA+nln\left(1-\alpha \right)$$

Repeating the method of plotting the straight line by $ln\frac{\beta \left(d\alpha /dT\right)}{exp\left(-E_{a}/RT\right)}$ versus ln(1−α), the slope of the line was the reaction order, and the intercept was lnA. In addition, for this study, if all of the activation values obtained by the Friedman methods were shown with a high coefficient of determinations at a variety of heating rates, Equation (14) was fit for the solution of the reaction order and the pre-exponential factor. Otherwise, two intercepts (lnA+lnf(α)) in Equation (13) obtained with a higher coefficient of determination at a certain conversation ratio would be applied, and two linear equations in two unknowns were simultaneously established for the solution of the reaction order and the pre-exponential factor.

#### 3.2.2. The Reasoning Process of Integral Methods

Flynn [38] confirmed that the *G*(α) in the integral method (Equation 8) could be rearranged by the temperature integral (P(μ)). The solution of P(μ) was shown in Appendix A.
(15)$$\left\{ \begin{array}{c} G\left(\alpha \right)=\frac{A}{\beta }\int_{T_{o}}^{T}e^{-E_{a}/RT}dT≅\frac{AE_{a}}{\beta R}\int_{\infty }^{\mu }-e^{-\mu }\mu^{-2}d\mu \\ =\frac{AE_{a}}{\beta R}P\left(\mu \right);\mu =\frac{E_{a}}{RT}\\ P\left(\mu \right)=\int_{\infty }^{\mu }-e^{-\mu }\mu^{-2}d\mu \Rightarrow \frac{e^{-\mu }}{\mu^{2}}\left(1-\frac{2!}{\mu }+\frac{3!}{\mu^{2}}-\frac{4!}{\mu^{3}}\cdots \right)\\ =\frac{e^{-\mu }}{\mu^{2}}\times \overset{\infty }{\underset{N=1}{\sum }}{\left(-1\right)}^{N-1}\frac{N!}{\mu^{N-1}};\;N\geq 1 \end{array}\right.$$
where *G*(α) is the integral equation of f(α), P(μ) is the temperature integral; and N is positive integer, which is greater or equal to one.

In the integral method, dα/dt disappeared, and the noise effect was avoided. However, the activation energy at certain heating rate equations (Equation (15)) was extremely intractable to acquire, which was attributed to P(μ). Thus, some methods were provided to simplify the solution of Equation (15). Akahira-Sunose [39] deduced that if the *N* in Equation (15) was equal to one, the *G*(α) could be simplified and expressed as:(16)$$\{\begin{array}{c} P\left(\mu \right)\approx \frac{e^{-\mu }}{\mu^{2}};N=1\\ G\left(\alpha \right)\approx \frac{AE_{a}}{\beta R}\times \frac{e^{-\mu }}{\mu^{2}}=\frac{T^{2}}{\beta }\frac{AR}{E}e^{-E_{a}/RT}\\ ln\frac{\beta }{T^{2}}=ln\frac{AR}{G\left(\alpha \right)E}-E_{a}/RT \end{array}$$

Equation (16) was similar to Kissinger approach (Equation (12)), and the activation energy could be solved by plotting $ln\frac{\beta }{T^{2}}$ VS. $\frac{1}{T}$. This type of integral method was utilized in the following sections and was labeled as the Kissinger-Akahira-Sunose method (abbreviated as the KAS (Kissinger-Akahira-Sunose) method) in the following studies. Similarly, if the *N* in Equation (15) was equal to two, combined with the Doyle approach [33], another type of integral method (Equation (17)) was acquired and labeled as the Flynn–Wall–Ozawa method (abbreviated as the FWO (Flynn–Wall–Ozawa) method) [40]. The slope ($-1.0516\;E_{a}/R$) of the straight line that was plotted by lnβ VS. $\frac{1}{T}$ was more convenient to solve the acquired energy:(17)$$\{\begin{array}{c} P\left(\mu \right)\approx \frac{e^{-\mu }}{\mu^{2}}\left(1-\frac{2!}{\mu }\right)\approx 0.00484e^{-1.0516\mu };N=2\\ \beta \approx \frac{AE}{RG\left(\alpha \right)}\times 0.00484e^{-1.0516\mu }\Rightarrow ln\beta \approx ln\frac{AE}{RG\left(\alpha \right)}-5.311-1.0516E/RT \end{array}$$

### 3.3. The Incineration Kinetic Analysis and Model of Stage One

No endothermic or exothermic phenomenon was obtained in Stage 1. Therefore, the TGA curves were utilized for the incineration kinetic analysis. Both dα/dT and *T* under various conversion rates $\alpha_{n}$ (*n* = 10%, 30%, 50%, 70%, and 90%) and heating rates $\beta_{n}$ (*n* = five K/min, 10 K/min, 15 K/min, 20 K/min, and 25 K/min) were shown in Table 4. Based on equations (13), (16), and (17), the results of linear fitting under the KAS method (Figure 3A), the Friedman method (Figure 3B), and the FWO method (Figure 3C) are shown in Figure 3. The activation energies under various conversion rates are shown in Table 5.

In Figure 3 and Table 4, when the conversion ratios are over 50% (Table 4), the corresponding coefficient of determinations under three methods were dramatically higher than 0.9555. Only 5% weight loss was detected in Stage 1 before $\alpha_{50\%}$, which was attributed to the dehydration process of oily sludge. When the conversion ratios exceed 50%, the volatilization (not combustion) process of volatiles was performed and confirmed in the thermal kinetic analysis of Stage 2 (Section 3.3). Compared with the coefficient of determinations obtained in the three methods, the Friedman method (Figure 3B) was more fitting for the kinetic modeling of volatilization. Furthermore, the activation energy changed with the conversion ratios in Stage 1, which means that neither the dehydration process nor the volatilization of volatiles was an elementary reaction [39,40]. For the volatilization process, the average activation energy ${\bar{E}}_{0\alpha -\left(50\%-100\%\right)}$ that was acquired under the Friedman method was 60.87 ± 5.27 KJ/mol. Then, the reaction orders and pre-exponential factors under various heating rates could be solved via Equation (14), as shown in Appendix A and Table 6. In addition, the average reaction order (or the average pre-exponential factor) was not the arithmetic mean value obtained at a variety of heating rates, but rather the slope (or intercept) of the straight line plotting by the $\bar{ln\frac{\beta n\left(d\alpha /dT\right)}{exp\left(-E_{a}/RT\right)}}$ versus ln(1−α).

The average reaction order was *n* = 0.82 ± 0.30, and the average pre-exponential factor was lnA = 13.32 ± 0.45. The volatilization kinetic model expressed in differential form and integral form were shown in Equation (18), respectively. Due to the simplification of the temperature integral (P(μ)), the differential form was better to state the volatilization kinetic model in Stage 1.
(18)$$d\alpha /dt=exp\left(13.32-60870/RT\right){\left(1-\alpha \right)}^{0.82};\;\alpha \in \left[0.5,\;1\right],T\in \left(435K,\;511K\right)$$

### 3.4. The Incineration Kinetic Analysis and Model of Stages Two, Three, and Four

Based on Figure 2C–E, the dα/dT and *T* values for stages two, three, and four under various conversion rates $\alpha_{n}$ (*n* = 10%, 30%, 50%, 70%, and 90%) and heating rates $\beta_{n}$ (*n* = five K/min, 10 K/min, 15 K/min, 20 K/min, and 25 K/min) are shown in Table 7.

The linear fitting results of stage two, three, and four under the KAS method (following Equation (13)), Friedman method (following Equation (16)), and FWO method (following Equation (17)) were shown in Figure 4, Figure 5 and Figure 6, respectively. The activation energies of each stage obtained under the three methods were shown in Table 8.

Both Stage 2 and Stage 3 were exothermic stages, and the $E_{0-\alpha n}$ (Table 8) values that were separately obtained by the KAS method, Friedman method, and FWO method at $\alpha_{n}$ (*n* = 10%, 30%, 50%, 70%, and 90%) were quite different. Meanwhile, $E_{0-\alpha n}$ apparently changed with $\alpha_{n}$ in each method. In Table 8, the average $\bar{E_{0}}$ of Stage 2 (Stage 3) obtained under the KAS method, Friedman method, and FWO method were 78.11 KJ/mol (98.82 KJ/mol), 58.07 KJ/mol (77.68 KJ/mol), and 65.63 KJ/mol (81.10 KJ/mol), respectively. In stages two and three (Table 8), the R^2^ at a variety of heating rates in the KAS method were all higher than that in the Friedman method and FWO method. Therefore, $\bar{E_{0}}=78.11\;\mathrm{KJ}/\mathrm{mol}$ and 98.82 KJ/mol were appropriate for the incineration kinetic modeling of Stage 2 and Stage 3, respectively.

In Stage 2, R^2^ was apparently changed with the heating rates, and relatively high R^2^ values by Friedman method (Table 8 and Figure 4B) were obtained at $\alpha_{10\%}$ (R^2^ = 0.9693) and $\alpha_{30\%}$ (R^2^ = 0.9774). The intercepts of $\alpha_{10\%}$ and $\alpha_{30\%}$ linear fitting curves (Figure 4B) plotted by $ln\left(\beta_{n}\times d\alpha /dT\right)$ versus $\frac{1}{T}$ were 19.33 ± 1.24 and 18.45 ± 1.39, respectively. Thus, two linear equations in two unknowns were simultaneously established and expressed in Equation (19):(19)$$\{\begin{array}{l} lnA+nln\left(1-\alpha_{10\%}\right)=19.33\pm 1.24\\ lnA+nln\left(1-\alpha_{30\%}\right)=18.45\pm 1.39 \end{array}$$

The reaction order and the pre-exponential factor for Stage 2 were lnA=19.69 and n=3.50, respectively. The reaction rate equation of Stage 2 was $f\left(\alpha \right)={\left(1-\alpha \right)}^{3.5}$. The kinetic model for the combustion of volatile components was expressed as:(20)$$d\alpha /dt=exp\left(19.69-78110/RT\right){\left(1-\alpha \right)}^{3.5};\alpha \in \left[0,\;1\right],\;T\in \left(511K,\;658K\right)$$

Similarly for Stage 3, relatively high R^2^ values (Table 8 and Figure 5B) were obtained at $\alpha_{10\%}$ and $\alpha_{50\%}$ by the Friedman method. The $ln\left(\beta_{n}\times d\alpha /dT\right)$ intercepts and R^2^ values of the $\alpha_{10\%}$ and $\alpha_{50\%}$ linear fitting curves (Figure 5B) were 20.74 ± 0.74, 0.9942 and 19.24 ± 1.32, 0.9602, respectively. The reaction order and the pre-exponential factor were solved from the followed equations:(21)$$\{\begin{array}{l} lnA+nln\left(1-\alpha_{10\%}\right)=20.74\pm 0.74\\ lnA+nln\left(1-\alpha_{50\%}\right)=19.24\pm 1.32 \end{array}$$

The reaction order, the pre-exponential factor, and the reaction rate equation for Stage 3 were lnA=21.00, n=2.50, and $f\left(\alpha \right)={\left(1-\alpha \right)}^{2.5}$, respectively. The kinetic model was expressed as:(22)$$d\alpha /dt=exp\left(21.00-98820/RT\right){\left(1-\alpha \right)}^{2.5};\;\alpha \in \left[0,\;1\right],\;T\in \left(658K,\;793K\right)$$

The endothermic phenomenon that existed in Stage 4 and the R^2^ of the linear fitting curves in the KAS method were dramatically higher than those in the Friedman method and FWO method. Therefore, $\bar{E_{0}}=15.96$ KJ/mol was the optimum parameter for the incineration kinetic modeling of Stage 4. As it was shown in Figure 6B, significant errors appeared in the linear fitting curves of Stage 4 under the Friedman method, and a relatively low R^2^ value was obtained in each heating rate. Thus, the reaction order and the pre-exponential factor could not be obtained in Equation (13) or Equation (14). We inferred that the probe that was utilized to detect the weight of the reactant was significantly affected by the operating temperatures and caused the apparent errors of dα/dt or dα/dT.

### 3.5. The Judgement of Sectionalized Modeling in Differential/Integral Method

For the incineration kinetics modeling of upstream oily sludge, the $E_{p}$ values of Stage 2 (or Stage 3, Table 1) obtained under the Kissinger approach were lower than the $\bar{E_{0}}$ values from the KAS method, which means that the peak operating temperature was more appropriate as the incineration temperature in engineering use. Attributed to the background noise or sensitivity of the probe, the activation energy ($E_{a}$) that was obtained in differential methods (the Friedman method) was imprecise, and the value of R^2^ also demonstrated that the integral method was more suitable than the differential method.

In comparison with the previous reports in Table 1, the differential method (Friedman method) was more convenient to obtain the pre-exponential factor (A) and the representation of the nth-order reaction rate equation f(α) or the reaction order (*n*). It seems that the comprehensive differential integral method was more reasonable to solve the basic three elements for the incineration kinetics analysis of upstream oily sludge. However, both the approximate solution of temperature integral P(μ) that existed in the integral method and the model that was utilized in engineering use should be evaluated and adjusted.

## 4. Conclusions

Based on whether the endothermic and/or exothermic reactions occurred (or not) in DSC tests, the TGA curves of upstream oily sludge could be sectionalized as four weight loss stages. No endothermic or exothermic reaction was detected, but a 10% weight loss ratio was obtained in Stage 1. The first and second exothermic reaction occurred in stages two (weight loss = 10%) and three (weight loss = 20%), and the ending temperatures were 614.62–664.91 K and 719.15–822.69 K, respectively. The temperature region of Stage 4 was between 635.968–900.00 °C, and the weight loss was 5%.

Five types of thermal reactions existed in the four incineration stages, i.e., dehydration and volatilization (Stage 1), the combustion of light components (Stage 2), the combustion of heavy components (Stage 3), and the decomposition of inorganic carbonate (Stage 4). The two combustion reactions caused the exothermic phenomenon, while the endothermic phenomenon was attributed to the decomposition reaction.

The integral methods (the FWO method and the KAS method) were efficient to obtain the activation energy, while the differential method (the Friedman method) was more suitable to solve the reaction order and pre-exponential factors. The average activation energies of stages one, two, three, and four were 60.87 KJ/mol, 78.11 KJ/mol, 98.82 KJ/mol, and 15.96 KJ/mol, respectively. The reaction order and pre-exponential factors of stages one, two, and three were 0.82, 3.50, and 2.50, and $e^{13.32}\min^{-1}$, $e^{19.69}\min^{-1}$, and $e^{21.00}\min^{-1}$, respectively. Due to the significant errors and relatively low R^2^ values that appeared in the linear fitting curves of Stage 4, the reaction order and the pre-exponential factor could not be obtained under the Friedman method.

The $E_{p}$ values of Stage 2 (or Stage 3, as shown in Table 1) obtained under the Kissinger approach were lower than the $\bar{E_{0}}$ values from the KAS method, which means that the peak operating temperature was more suitable as the incineration temperature in engineering use.

## Figures and Tables

**Figure 1 ijerph-16-00384-f001:**
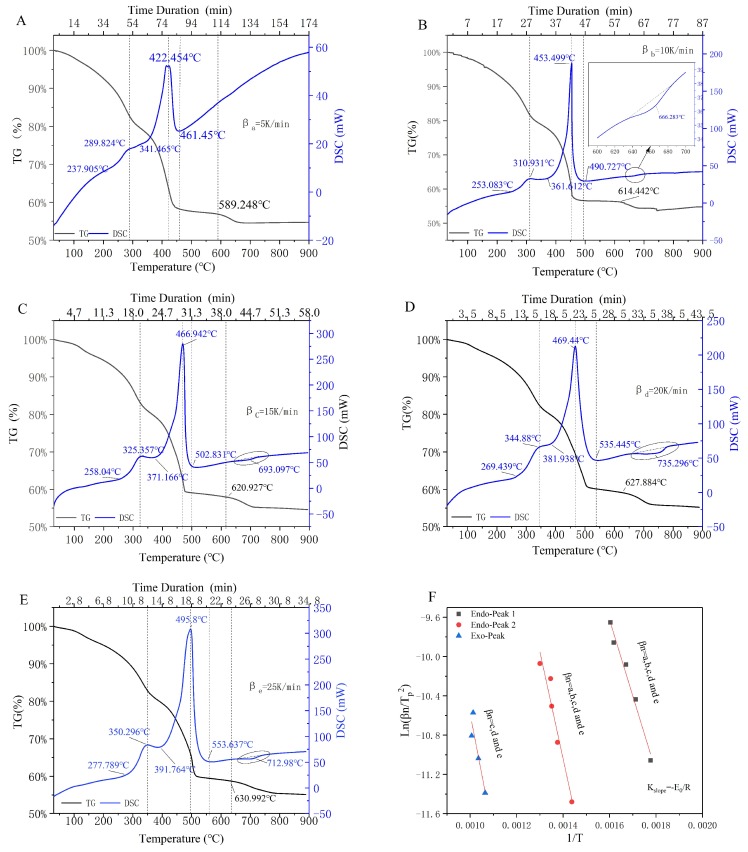
The DSC/TGA analysis of upstream oily sludge at $\beta_{n}$ (*n* = a, b, c, d, and e); $\beta_{a}=$ 5 K/min (**A**), $\beta_{b}=$ 10 K/min (**B**), $\beta_{c}=$ 15 K/min (**C**), $\beta_{c}=$ 20 K/min (**D**) and $\beta_{d}=$ 25 K/min (**E**), endothermic/exothermic peak analysis (**F**).

**Figure 2 ijerph-16-00384-f002:**
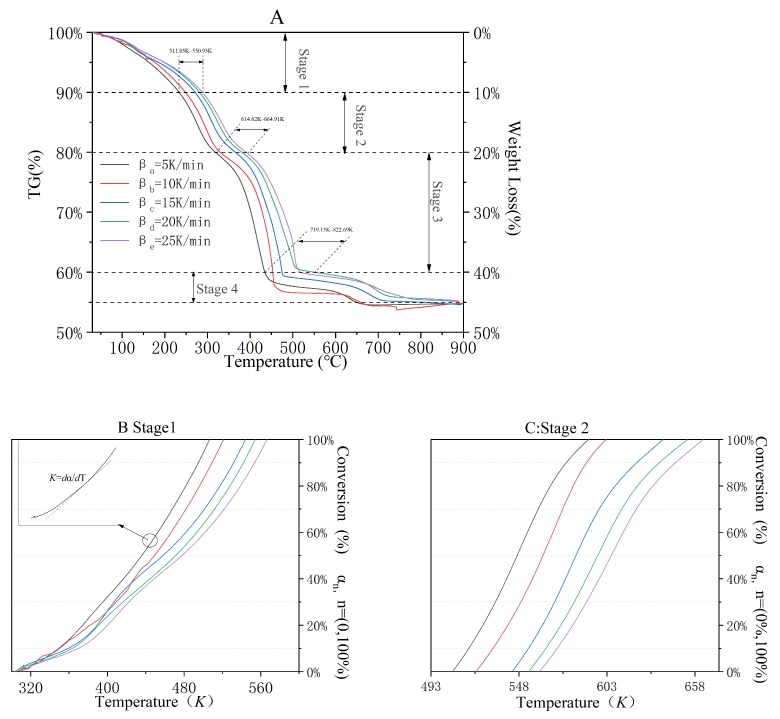
Four sectionalized stages in TGA. TGA test carried out in different heating rates (**A**); TGA test of stage 1 (**B**); TGA test of stage 2 (**C**); TGA test of stage 3 (**D**); TGA test of stage 4 (**E**).

**Figure 3 ijerph-16-00384-f003:**
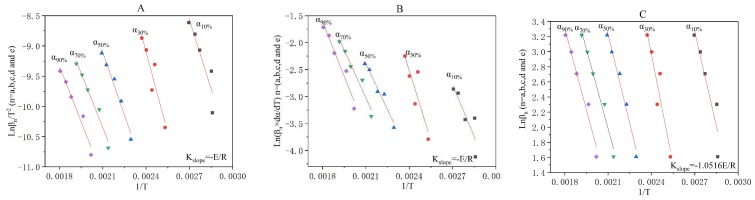
The linear fitting results for Stage 1 under the KAS, Friedman, and FWO methods ((**A**)—KAS method, (**B**)—Friedman method, and (**C**)—FWO method).

**Figure 4 ijerph-16-00384-f004:**
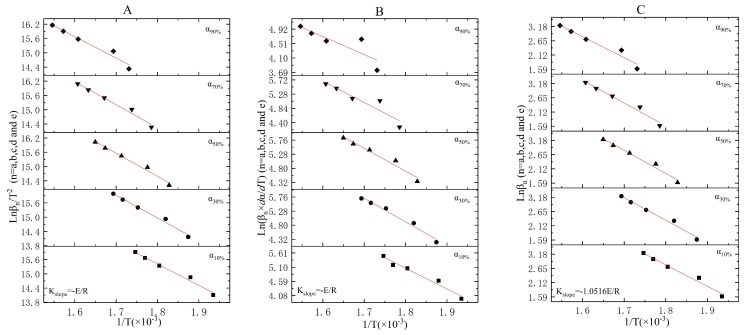
The linear fitting results for Stage 2 under the KAS, Friedman, and FWO methods ((**A**)—KAS method, (**B**)—Friedman method, and (**C**)—FWO method).

**Figure 5 ijerph-16-00384-f005:**
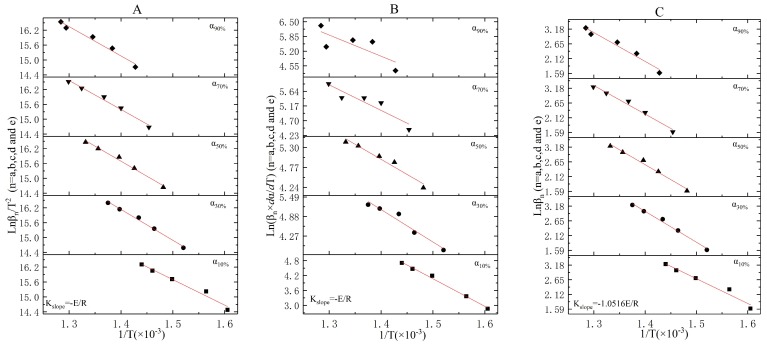
The linear fitting results for Stage 3 under the KAS, Friedman, and FWO methods ((**A**)—KAS method, (**B**)—Friedman method, and (**C**)—FWO method).

**Figure 6 ijerph-16-00384-f006:**
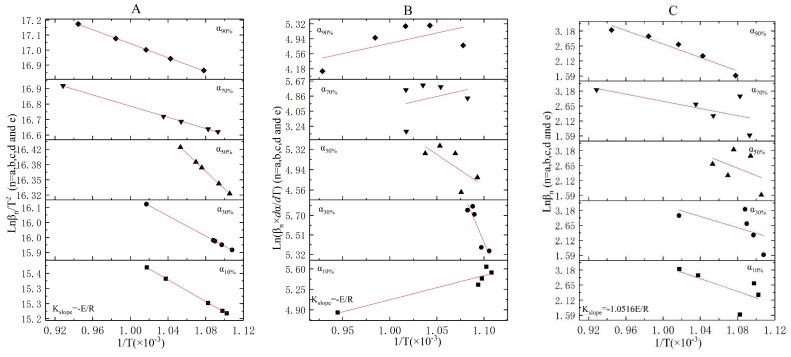
The linear fitting results for Stage 4 under the KAS, Friedman, and FWO methods ((**A**)—KAS method, (**B**)—Friedman method, and (**C**)—FWO method).

**Table 1 ijerph-16-00384-t001:** Thermal kinetics analysis and modeling methods reported in the previous studies.

Materials	Atmosphere	Thermal Test Method	Modeling Method	Basic Three Elements	Reference
Oil sludge, Phenolic plastic	Nitrogen	TGA/DTG	Integral	$E_{a}$	[11,34]
Oil sludge	Nitrogen	TGA	Differential	$E_{a}$, A and *n*	[12]
Polyurethane Foams	Nitrogen	TGA	Differential	$E_{a}$	[30,31]
Chalcogenide Ge_2_Sb_2_Te_5_	Nitrogen	DSC	Integral	$E_{a}$	[36]
Polyurethane	Nitrogen	TGA	Differential	$E_{a}$, A and *n*	[37]
Rice husk	Nitrogen	TGA	Integral	$E_{a}$	[39]

TGA: thermogravimetric analysis; DTG: derivative thermogravimetric analysis; DSC: differential scanning calorimetry.

**Table 2 ijerph-16-00384-t002:** Features of upstream oily sludge.

Parameters	Heating Values (MJ/Kg, dry basis)	Ash Content (wt.%, dry basis)	Moisture (wt.%)	Bulk Density (kg/m³)
Values	35.45 ± 3.64	57.56 ± 1.23	3.24 ± 1.08	1366.25 ± 46.73

**Table 3 ijerph-16-00384-t003:** Thermal characteristics of upstream oily sludge in stages one through four.

Parameters		Stage 1	Stage 2	Stage 3	Stage 4
DSC	Endo/Exo ^1^	ND ^2^	Exo	Exo	Endo
TGA	Weight Loss	10%	10%	20%	5%
Peak temperature at $\beta_{n}$ (K)	$\beta_{a}=5\;K/\min$	ND	562.974	695.604	ND
	$\beta_{b}=10\;K/\min$	ND	584.081	726.649	939.433
	$\beta_{c}=15\;K/\min$	ND	598.507	740.092	965.624
	$\beta_{d}=20\;K/\min$	ND	618.030	742.590	993.691
	$\beta_{e}=25\;K/\min$	ND	623.446	768.950	987.380
*E_p_* (KJ/mol)	$ln\left(\beta_{n}/T_{p}{}^{2}\right)–1/T_{p}$	ND	64.43 ± 5.34	90.71 ± 13.35	102.11 ± 28.93
	R^2^	ND	0.9798	0.9389	0.8616

Note: 1. Endothermic (Endo)/Exothermic (Exo); 2. Not Detected.

**Table 4 ijerph-16-00384-t004:** Parameters for incineration kinetic modeling in Stage 1.

$\alpha_{n}$	$\beta_{a}=5\;K/\min$		$\beta_{b}=10\;K/\min$		$\beta_{c}=15\;K/\min$		$\beta_{d}=20\;K/\min$		$\beta_{e}=25\;K/\min$	
	dα/dT (×10^−3^)	T (K)	dα/dT (×10^−3^)	T (K)	dα/dT (×10^−3^)	T (K)	dα/dT (×10^−3^)	T (K)	dα/dT (×10^−3^)	T (K)
$\alpha_{10\%}$	3.26	349.781	3.33	350.605	2.16	358.763	2.65	365.125	2.29	369.640
$\alpha_{30\%}$	4.51	394.986	4.35	409.822	5.24	406.490	3.64	416.323	4.22	421.908
$\alpha_{50\%}$	5.59	435.879	5.19	448.973	3.63	458.567	4.09	470.557	3.66	477.807
$\alpha_{70\%}$	6.90	467.820	6.75	481.817	5.78	501.161	5.77	511.133	5.51	521.753
$\alpha_{90\%}$	7.97	495.421	7.99	509.078	7.42	531.442	7.73	541.839	7.19	553.763

**Table 5 ijerph-16-00384-t005:** The activation energies of Stage 1 obtained under the KAS, FWO, and Friedman methods.

$\alpha_{n}$ $\left(\beta_{a}–\beta_{e}\right)$	KAS Method $\ln\left(\beta_{n}/T^{2}\right)–1/T$		FWO Method $\ln\left(\beta_{n}\right)–1/T$		Firedman Method $\ln\left(\beta_{n}\times d\alpha /dT\right)–1/T$	
	$E_{0-\alpha n}(\mathrm{KJ}/\mathrm{mol})$	R^2^	$E_{0-\alpha n}(\mathrm{KJ}/\mathrm{mol})$	R^2^	$E_{0-\alpha n}(\mathrm{KJ}/\mathrm{mol})$	R^2^
$\alpha_{10\%}$	59.26 ± 15.62	0.8453	65.84 ± 14.79	0.8676	53.56 ± 17.26	0.7572
$\alpha_{30\%}$	56.83 ± 17.05	0.8622	63.51 ± 16.13	0.8818	45.41 ± 23.09	0.7633
$\alpha_{50\%}$	54.53 ± 5.44	0.9738	61.93 ± 5.11	0.9798	47.32 ± 5.83	0.9571
$\alpha_{70\%}$	54.04 ± 5.15	0.9691	60.45 ± 4.83	0.9774	49.06 ± 9.38	0.9558
$\alpha_{90\%}$	53.78 ± 6.24	0.9565	60.24 ± 5.87	0.9685	55.80 ± 7.08	0.9555

**Table 6 ijerph-16-00384-t006:** The reaction order and pre-exponential factor of Stage 1 at a variety of heating rates.

Parameters	Reaction Order (*n*)	Pre-Exponential Factor (*ln A*)	Coefficient of Determination (R^2^)
β=5 K/min	0.97 ± 0.28	13.72 ± 0.44	0.9236
β=10 K/min	0.87 ± 0.25	13.79 ± 0.40	0.9202
β=15 K/min	0.85 ± 0.29	13.46 ± 0.45	0.9012
β=20 K/min	0.81 ± 0.30	13.42 ± 0.47	0.8766
β=25 K/min	0.82 ± 0.30	13.32 ± 0.45	0.8868
Average	0.82 ± 0.30	13.32 ± 0.45	0.9023

**Table 7 ijerph-16-00384-t007:** Parameters for incineration kinetic modeling in stages two, three, and four.

$\alpha_{n}$	Stage	$\beta_{a}=5\;K/\min$		$\beta_{b}=10\;K/\min$		$\beta_{c}=15\;K/\min$		$\beta_{d}=20\;K/\min$		$\beta_{e}=25\;K/\min$	
		dα/dT (×10^−3^)	T (*K*)	dα/dT (×10^−3^)	T (*K*)	dα/dT (×10^−3^)	T (*K*)	dα/dT (×10^−3^)	T (*K*)	dα/dT (×10^−3^)	T (*K*)
$\alpha_{10\%}$	Stage 2	10.57	516.785	10.08	531.892	10.49	554.185	8.89	565.030	9.81	572.626
	Stage 3	3.51	622.887	2.89	639.315	4.39	667.270	4.36	684.588	4.43	694.448
	Stage 4	50.82	902.590	21.16	904.648	8.05	914.979	5.77	923.805	4.72	927.736
$\alpha_{30\%}$	Stage 2	13.60	533.479	13.03	549.453	14.35	570.510	12.99	582.810	12.09	590.517
	Stage 3	9.25	657.685	7.96	682.952	9.50	697.241	8.41	715.626	7.60	727.265
	Stage 4	56.23	907.129	21.91	911.520	12.60	934.999	10.61	948.819	7.82	959.173
$\alpha_{50\%}$	Stage 2	15.53	546.946	15.83	563.042	15.21	583.725	13.97	597.489	13.73	606.171
	Stage 3	13.73	674.935	13.56	701.422	10.61	716.113	10.48	737.392	9.32	750.831
	Stage 4	45.91	911.038	30.19	917.644	14.53	949.547	11.59	965.994	7.69	983.627
$\alpha_{70\%}$	Stage 2	14.17	560.069	15.98	575.404	11.44	598.262	11.76	612.476	10.76	622.225
	Stage 3	16.51	687.783	19.34	714.310	15.14	731.354	11.44	755.153	14.27	769.902
	Stage 4	41.20	914.145	31.44	923.775	12.63	963.750	9.04	983.212	5.73	1015.759
$\alpha_{90\%}$	Stage 2	8.51	577.680	10.3	590.512	6.53	621.731	6.09	635.915	5.94	646.828
	Stage 3	15.38	700.390	27.47	722.863	19.72	743.134	11.03	772.596	22.36	778.701
	Stage 4	25.62	919.106	32.64	929.565	6.11	982.866	0.97	1076.961	2.45	1058.503

**Table 8 ijerph-16-00384-t008:** The activation energies of stages two, three, and four obtained from the KAS, FWO, and Friedman methods.

Stage	$\alpha_{n}$ $\left(\beta_{a}–\beta_{e}\right)$	KAS Method $\ln\left(\beta_{n}/T^{2}\right)–1/T$		Friedman Method $\ln\left(\beta_{n}\times d\alpha /dT\right)–1/T$		FWO Method $\ln\left(\beta_{n}\right)–1/T$	
		$E_{0-\alpha n}\;(\mathrm{KJ}/\mathrm{mol})$	R^2^	$E_{0-\alpha n}\;(\mathrm{KJ}/\mathrm{mol})$	R^2^	$E_{0-\alpha n}\;(\mathrm{KJ}/\mathrm{mol})$	R^2^
Stage 2	$\alpha_{10\%}$	75.40 ± 5.81	0.9825	61.80 ± 6.34	0.9693	62.84 ± 5.53	0.9772
	$\alpha_{30\%}$	79.10 ± 5.56	0.9854	66.58 ± 5.84	0.9774	69.77 ± 5.31	0.9810
	$\alpha_{50\%}$	80.59 ± 5.88	0.9843	64.64 ± 7.59	0.9602	67.25 ± 5.62	0.9795
	$\alpha_{70\%}$	80.29 ± 6.81	0.9789	54.68 ± 11.97	0.8743	66.74 ± 6.50	0.9723
	$\alpha_{90\%}$	75.16 ± 8.69	0.9614	42.63 ± 15.31	0.7210	61.57 ± 8.26	0.9487
Stage 3	$\alpha_{10\%}$	94.42 ± 8.09	0.9738	92.41 ± 4.07	0.9942	70.83 ± 7.71	0.9654
	$\alpha_{30\%}$	103.31 ± 5.03	0.9930	84.01 ± 10.33	0.9565	87.32 ± 4.85	0.9908
	$\alpha_{50\%}$	100.83 ± 6.18	0.9888	66.66 ± 7.83	0.9602	84.63 ± 5.94	0.9853
	$\alpha_{70\%}$	97.26 ± 6.77	0.9857	67.74 ± 15.01	0.8703	80.98 ± 6.52	0.9808
	$\alpha_{90\%}$	98.28 ± 9.40	0.9732	77.59 ± 35.79	0.6103	81.76 ± 8.99	0.9647
Stage 4	$\alpha_{10\%}$	15.68 ± 0.074	0.9999	−34.35 ± 6.88	0.8925	89.05 ± 56.65	0.4516
	$\alpha_{30\%}$	15.71 ± 0.073	0.9999	179.03 ± 45.85	0.8355	78.65 ± 66.51	0.3179
	$\alpha_{50\%}$	15.40 ± 0.068	0.9999	92.93 ± 60.19	0.4687	103.69 ± 127.91	0.1797
	$\alpha_{70\%}$	16.54 ± 0.197	0.9996	95.77 ± 170.19	0.0955	50.20 ± 33.35	0.4303
	$\alpha_{90\%}$	16.46 ± 0.195	0.9996	−43.57±31.13	0.3949	94.35 ± 14.70	0.9321

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
