# Peer review of "Incineration Kinetic Analysis of Upstream Oily Sludge and Sectionalized Modeling in Differential/Integral Method"

_ijerph, 2019, doi:10.3390/ijerph16030384_

Round 1
Reviewer 1 Report
In this paper, multiple non-isothermal thermogravimetric analysis (TGA) and differential scanning calorimetry (DSC) were applied for the kinetic analysis of upstream oily sludge in air conditions. The topic is scientific and practical, and the experimental design is logical. Therefore, I suggest minor revision for the manuscript before publishing if the authors can solve the following questions and revise the manuscript carefully.
1. Introduction
The difference from previous studies should be described in detail to highlight novelty and significance of this manuscript.
2. Material and methods
Section 2.1, the features analysis of upstream oily sludge employed in this study should be placed in section “results and discussion”, if this work was accomplished in this study.
Section 2.1 and 2.2 used the same titles, please check them.
3. Results and discussion
The fonts in all figures are too small for readers, please adjust them.
The effective numbers for all the data present in this paper should be consistent.
4. Please revise the manuscript carefully to improve the English level.
5. Please check the references in terms of requirement of the journal.
Author Response
Response to Reviewers’ Comments
Dear reviewer and editor:
We are very grateful to the reviewer’s constructive and elaborate comments. We will try our best to improve our paper, and thank you for your sincere instruction! According with your advice and request, we amended the relevant part and made extensive modification in the formats of PDF. The changes in the revised paper have been highlighted in red. All the responses were summarized for the editor.
Thanks again for the constructive comments!
Corresponding author: Qi yuanfeng
School of environmental and municipal engineering,
Qingdao University of Technology
Response to Reviewer 1 Comments
Thanks for the constructive comments. We have revised the manuscript as suggested. The changes in the revised paper have been high-lighted in red. Responses are listed after the comments as follows:
Comment 1: Introduction, the difference from previous studies should be described in detail to highlight novelty and significance of this manuscript.
Response 1: thank you for your comment, a new Table 1 was add in section 1 to introduce the previous studies for thermal kinetics analysis methods, and a new section 3.5 was added to highlight not only the novelty and significance but also the limitation of this manuscript.
Comment 2: Material and methods, Section 2.1, the features analysis of upstream oily sludge employed in this study should be placed in section “results and discussion”, if this work was accomplished in this study. Section 2.1 and 2.2 used the same titles, please check them.
Response 2: this part has be rearranged and placed in section 3.1.1.
Comment 3: Results and discussion, the fonts in all figures are too small for readers, please adjust them. The effective numbers for all the data present in this paper should be consistent.
Response 3: the fonts in all figures and effective numbers for all the data have been adjusted.
Comment 4: Please revise the manuscript carefully to improve the English level. And comment 5. Please check the references in terms of requirement of the journal.
Response 4: Thanks for the sincere suggestion and comments, and we will try our best to improve our paper. The language of the uploaded manuscript was revised.
Finally, we thank the reviewer to give constructive comments to improve our manuscript. Please do not hesitate to contact us if you have any queries.

Reviewer 2 Report
In this study, the oily sludge generated from the oil production industry and obtained from the bottom of storage oil tanks was employed as the target hazardous waste. DSC/TG test was carried out for the thermal kinetic analysis and categorized as four different sections. Four classic thermal kinetic methods were successfully utilized for the modeling of each section. As a whole, depend on the endothermal or exothermal phenomenon in DSC curves, the categorized method to divide the TG curves into four sections or stages is reasonable. For the complex solid waste, the mentioned differential/integral modeling method is interesting. However, the manuscript should be improved by the authors before publication, and I recommend major revision as follows:
1: In introduction, the role of incineration kinetic analysis or modeling for engineering use should be introduced. Additionally, the novelty, significance of this study and its importance for real projects should be also stated in detail.
2: Table 1 should be placed in results and discussion. And the testing methods should be introduced in detail.
3: Line 228, equation 12. Tn-p should be introduced, whether it is changed with β or βn?
4: Line 311. The coefficient of determination should be marked as R2, some superscripts were incorrectly labeled as R2 in section 3.4. Gas constant was R, which will cause different meanings for readers.
5: Line 304-305. Figure 4 and Table 5 represent the same massage, it is better to place Figure 4 in supplementary materials.
6: Line 310, equation 18. The equation in the differential form was enough for the incineration kinetic model, the integral form was not necessary.
7:Line 369. Conclusions should be abbreviated and rewritten.
8: The authors should check the English expression to avoid any grammar and spelling mistakes. Additionally, the references should also be carefully revised by the authors according to the guide for authors of the journal.
Author Response
Response to Reviewer 2 Comments
Thanks for the constructive comments. We have revised the manuscript as suggested. The changes in the revised paper have been high-lighted in red. Responses are listed after the comments as follows:
Comment1:In introduction, the role of incineration kinetic analysis or modeling for engineering use should be introduced. Additionally, the novelty, significance of this study and its importance for real projects should be also stated in detail.
Response 1: we truly thank you for your constructive comment to give this study with a perfect beginning and ending. We do think this comparison in a table was necessary. Based on your suggestion, a new Table 1 was add in “introduction” to report the progress of other researches, and a new section 3.5 was established to describe the use of the proposed thermal modeling method. Advantages and Limitations was discussed as well.
Comment2: Table 1 should be placed in results and discussion. And the testing methods should be introduced in detail.
Response 2: this part has be placed in section 3.1.1, and the quondam Table 1 has been labeled as Table 2 in the uploaded manuscript. The pretreatment and apparatus of upstream oily sludge for the heating value and ash content test were introduced. The test of bulk density and moisture content for undried upstream oily sludge was detailed introduced and reported in our previous studies [27,29].and the operation process of TGA/DSC test was revised
Comment3:Line 228, equation 12. Tn-p should be introduced, whether it is changed with β or βn?
Response 3: Where T_(n-p) was the peak temperature obtained in DSC curve, it was changed with β_n.
Comment4 Line 311. The coefficient of determination should be marked as R2, some superscripts were incorrectly labeled as R2 in section 3.4. Gas constant was R, which will cause different meanings for readers.
Response 4: the superscript for all the coefficient of determinations has been checked, and the wrong numbered coefficient of determinations has been revised.
Comment5: Line 304-305. Figure 4 and Table 5 represent the same massage, it is better to place Figure 4 in supplementary materials.
Response 5:Thank you for your comments , figure 4 has been placed in supplementary materials.
Comment6:Line 310, equation 18. The equation in the differential form was enough for the incineration kinetic model, the integral form was not necessary.
Response 6: based on your comment, only differential form was utilized for the incineration kinetic model, the integral form of incineration kinetic model has been deleted.
Comment 7: Line 369. Conclusions should be abbreviated and rewritten.
Response 7: thank you for your comments, Conclusions has been abbreviated and revised.
Comment 8:The authors should check the English expression to avoid any grammar and spelling mistakes. Additionally, the references should also be carefully revised by the authors according to the guide for authors of the journal.
Response 8: Thanks for the sincere suggestion and comments, and we will try our best to improve our paper. The language of the uploaded manuscript was revised.
Finally, we thank the reviewer to give constructive comments to improve our manuscript. Please do not hesitate to contact us if you have any queries.

Reviewer 3 Report
The paper “Incineration Kinetic Analysis of Upstream Oily Sludge and Sectionalized Modeling in Differential/Integral Method”, authors Yanqing Zhang, Xiaohui Wang, Yuanfeng Qi, and Fei Xi
The manuscript Can Be published in International Journal of Environmental Research and Public Health– after MODERATE REVISION.
In this study the authors present a multiple non-isothermal thermogravimetric analysis (TGA) and differential scanning calorimetry (DSC) for establish the kinetic analysis of upstream oily sludge in air conditions. A differential-integral method to obtain the incineration kinetic model was provided. The integral method was used to obtain the average activation energy of each four stage, and the differential method was suitable to gain the nth-order reaction rate equation and pre-exponential factor.
Observation:
1. The topic is of general interest, and the presentation is clear
2. This article contains new aspects regarding the incineration of upstream oily sludge.
3. In manuscript the authors must underline the major findings of their work and explain how the use of their proposed analysis represents a progress comparatively with other researches. The table with comparison is necessary.
4. The Abstract is OK.
5. The key words permit found article in the current registers or indexes.
6. The Introduction reflects state of the art.
7. The text is easy to understand by scientists in other disciplines.
8. The text is presented and arranged clearly and concisely.
9. Please verify “Differ from the downstream oily sludge discharged in the refining process, the upstream oily sludge principally generated in the process of crude oil storage.”
10. Please verify 2.2. Materials and agents
11. Please verify after eq. 8 “?0 is the initial and final operating temperature.”
12. Please complete acknowledgement.
13. The figures haven’t good quality. The axes aren’t clear. Please verify all figures.
14. The Conclusion is OK.
15. Please verify the journal for reference [16] E. Zubaidy, D. Abouelnasr, Fuel recovery from waste oily sludge using solvent extraction, Process. Saf. Environ., 2010,88,318-326, because I find Process Safety and Environmental Protection.
16. Please verify journal abbreviation and respect author guide.
17. I recommend checking the English in manuscript.
Author Response
Response to Reviewer 3 Comments
Thanks for the reviewer’s constructive and elaborate comments. We’d very much appreciate your advice. And show ours respect to your strict attitude towards the academic thesis. We are pleased to learn a lot from your suggestions. The revised sentences in the manuscript have been colored red.
Comment 1,2,4-8 and 14
1. The topic is of general interest, and the presentation is clear
2. This article contains new aspects regarding the incineration of upstream oily sludge.
4. The Abstract is OK.
6. The Introduction reflects state of the art.
7. The text is easy to understand by scientists in other disciplines.
8. The text is presented and arranged clearly and concisely.
14. The Conclusion is OK.
Response 1, 2,4,6-8 and 14: It’s my great honor and we are truly thank you for your positive affirmation on this study! We will make arduous efforts in the following works and studies.
Comment 3 In manuscript the authors must underline the major findings of their work and explain how the use of their proposed analysis represents a progress comparatively with other researches. The table with comparison is necessary.
Response 3: we truly thank you for your constructive comment to give this study with a perfect beginning and ending. We do think this comparison in a table was necessary. Based on your suggestion, a new Table 1 was add in “introduction” to report the progress of other researches, and a new section 3.5 was established to describe the use of the proposed thermal modeling method. Advantages and Limitations was discussed as well.
Comment 9 Please verify “Differ from the downstream oily sludge discharged in the refining process, the upstream oily sludge principally generated in the process of crude oil storage.”
Response 9: this sentence has been rewritten as: “Upstream oily sludge is the most significant solid waste generated in the oil production industry and mainly discharged from the crude oil storage process”
Comment 10 Please verify 2.2. Materials and agents
Response 10: section 2, including “materials and reagents” and “apparatus and methods” has been rewritten. Features of upstream oily sludge was shown in section 3.1.1.
Comment 11 Please verify after eq. 8 “?0 is the initial and final operating temperature.”
Response 11: “?0 is the initial and final operating temperature.” Has been revised as : ?0 and T is the initial and final operating temperature, respectively
Comment 12 Please complete acknowledgement.
Response12 acknowledgement was added
Comment 13 The figures haven’t good quality. The axes aren’t clear. Please verify all figures.
Response13 all the fonts and lables of axes in all figures have been revised and enlarged.
Comment 15 -16
15 Please verify the journal for reference [16] E. Zubaidy, D. Abouelnasr, Fuel recovery from waste oily sludge using solvent extraction, Process. Saf. Environ., 2010,88,318-326, because I find Process Safety and Environmental Protection.
16 Please verify journal abbreviation and respect author guide.
Response 15-16: :we truly thank you for your strict attitude towards the academic thesis, the referenced report- [16] was published in Process Safety and Environmental Protection, and the abbreviated title of this journal was Process Saf. Environ. Prot. All the abbreviated title of referenced journals has been checked and revised.
Comment 17 I recommend checking the English in manuscript.
Response17 Thanks for the sincere suggestion and comments, and we will try our best to improve our paper. The language of the uploaded manuscript was revised.
Finally, we thank the reviewer to give constructive comments to improve our manuscript. Please do not hesitate to contact us if you have any queries.

Reviewer 4 Report
the article is well structured and evidenced hard work, however, the english need to be improved. and the size of the figures and their legends and axes are really difficult to see.
Author Response
Response to Reviewer 4 Comments
Thanks for the constructive comments, and we will try our best to improve our paper, and thank you for your sincere instruction!
Comment: the article is well structured and evidenced hard work, however, the english need to be improved. and the size of the figures and their legends and axes are really difficult to see
Response : It’s my great honor and we are truly thank you for your positive affirmation on this study! We will make arduous efforts in the following works and studies. All the fonts and lables of axes in all figures have been revised and enlarged. And we will try our best to improve our paper. The language of the uploaded manuscript was revised.
Finally, we thank the reviewer to give constructive comments to improve our manuscript. Please do not hesitate to contact us if you have any queries.

Round 2
Reviewer 2 Report
Dear Editor,
The manuscript has been carefully revised by the authors according to the reviewers' comments, and I suggest acceptance for the revised manuscript.
Reviewer 4 Report
accept